# Mucosal Genes Encoding Clock, Inflammation and Their Mutual Regulators Are Disrupted in Pediatric Patients with Active Ulcerative Colitis

**DOI:** 10.3390/ijms25031488

**Published:** 2024-01-25

**Authors:** Sapir Labes, Oren Froy, Yuval Tabach, Raanan Shamir, Dror S. Shouval, Yael Weintraub

**Affiliations:** 1Department of Developmental Biology and Cancer Research, Institute for Medical Research Israel-Canada, The Hebrew University of Jerusalem, Jerusalem 91905, Israel; sapir.labes@mail.huji.ac.il (S.L.); tabachy@gmail.com (Y.T.); 2Institute of Biochemistry, Food Science and Nutrition, Robert H Smith Faculty of Agriculture, Food and Environment, The Hebrew University, Rehovot 7610001, Israel; 3Institute of Gastroenterology, Nutrition and Liver Diseases, Schneider Children’s Medical Center of Israel, Petach Tikva 4920235, Israel; raanan@shamirmd.com (R.S.); dror.shouval@gmail.com (D.S.S.); dryaelw@gmail.com (Y.W.); 4Faculty of Medicine, Tel Aviv University, Tel Aviv 69978, Israel

**Keywords:** UC, circadian rhythms, clock gene expression, inflammation, mutual regulators

## Abstract

Patients with active ulcerative colitis (UC) display a misalignment of the circadian clock, which plays a vital role in various immune functions. Our aim was to characterize the expression of clock and inflammation genes, and their mutual regulatory genes in treatment-naïve pediatric patients with UC. Using the Inflammatory Bowel Disease Transcriptome and Metatranscriptome Meta-Analysis (IBD TaMMA) platform and R algorithms, we analyzed rectal biopsy transcriptomic data from two cohorts (206 patients with UC vs. 20 healthy controls from the GSE-109142 study, and 43 patients with UC vs. 55 healthy controls from the GSE-117993 study). We compared gene expression levels and correlation of clock genes (*BMAL1*, *CLOCK*, *PER1*, *PER2*, *CRY1*, *CRY2*), inflammatory genes (*IκB*, *IL10*, *NFκB1*, *NFκB2*, *IL6*, *TNFα*) and their mutual regulatory genes (*RORα*, *RORγ*, *REV-ERBα*, *PGC1α*, *PPARα*, *PPARγ*, *AMPK*, *SIRT1*) in patients with active UC and healthy controls. The clock genes *BMAL1*, *CLOCK*, *PER1* and *CRY1* and the inflammatory genes *IκB*, *IL10*, *NFκB1*, *NFκB2*, *IL6* and *TNFα* were significantly upregulated in patients with active UC. The genes encoding the mutual regulators *RORα*, *RORγ*, *PGC1α*, *PPARα* and *PPARγ* were significantly downregulated in patients with UC. A uniform pattern of gene expression was found in healthy controls compared to the highly variable expression pattern in patients with UC. Among the healthy controls, inflammatory genes were positively correlated with clock genes and they all showed reduced expression. The difference in gene expression levels was associated with disease severity and endoscopic score but not with histological score. In patients with active UC, clock gene disruption is associated with abnormal mucosal immune response. Disrupted expression of genes encoding clock, inflammation and their mutual regulators together may play a role in active UC.

## 1. Introduction

Ulcerative colitis (UC) is a chronic inflammatory disease affecting the colon, and its incidence is rising worldwide. The pathogenesis of UC is multifactorial, involving genetic predisposition, epithelial barrier defects, dysbiosis, dysregulated immune responses and environmental factors [1]. It is characterized by relapsing and remitting symptoms due to mucosal inflammation, starting at the rectum and extending to proximal segments of the colon [1]. It has been shown that circadian clock gene levels are reduced in both intestinal biopsies and peripheral blood cells of patients with inflammatory bowel disease (IBD) and specifically UC [2,3,4,5].

The circadian clock is an internal oscillator that controls physiological and behavioral processes in living organisms around the clock [6]. The circadian clock is hierarchical. In mammals, the central oscillator is located in the hypothalamic suprachiasmatic nucleus (SCN) and acts as the master pacemaker. This pacemaker synchronizes and entrains peripheral clocks, which are found in most cells throughout the body. The clock comprises transcriptional–translational autoregulatory feedback loops. The core clock mechanism consists of the proteins CLOCK and BMAL1, which are the transcriptional activators, and PER1, PER2, CRY1 and CRY2, which are the transcriptional repressors [7]. 

The circadian clock regulates the activity of the immune system. This regulation involves changes around the circadian cycle in antibody production, complement levels, the number of circulating lymphocytes, natural killer cells, cytokine production, host–pathogen interactions and the activation of innate and adaptive immunity [8,9,10]. At the molecular level, this is achieved by key clock regulatory proteins that have a dual function as regulators of the expression of certain pro- and anti-inflammatory factors. For example, REV-ERBα and retinoic acid receptor-related orphan receptor α (RORα), core clock regulatory proteins which regulate BMAL1 expression [11,12], also control the expression of inflammatory cytokines through the NF-κB signaling pathway [13,14,15,16]. Similarly, peroxisome proliferator-activated receptor α (PPARα), PPARγ and PPARγ transcriptional co-activator 1α (PGC1α) regulate BMAL1 expression [17,18] as well as suppress pro-inflammatory gene expression [19,20,21,22]. Finally, adenosine monophosphate-activated protein kinase (AMPK) and Sirtuin 1 (SIRT1) downregulate the expression of PERs and CRYs, core clock genes [23,24,25,26,27,28], as well as downregulate the expression of pro-inflammatory cytokines [29]. Thus, the emerging data support the notion that clock disruption is associated with inflammation, and that regulators of the circadian clock may play a role in the inflammatory response [10,30]. Herein, we aimed to investigate circadian clock and inflammatory gene expression as well as the expression of genes encoding their mutual regulators in rectal biopsies of pediatric patients with active UC, upon diagnosis, prior to medical treatment.

## 2. Results

### 2.1. Expression Level of Genes Encoding Clock, Inflammation and Their Mutual Regulators

To evaluate circadian disruption in pediatric patients with UC compared to healthy controls, we analyzed the expression of the aforementioned genes in two different transcriptomic studies. In the GSE-109142 study, clock gene expression analysis revealed that *BMAL1*, *CLOCK*, *PER1* and *CRY1* were significantly upregulated in patients with UC (*p* < 0.0001, *p* < 0.01, *p* < 0.05, *p* < 0.0001 and *p* < 0.0001, respectively) compared to healthy controls (Figure 1A), whereas PER2 was significantly reduced (*p* < 0.05) (Figure 1A). These findings portray a disruption in the core clock oscillator in rectal biopsies of patients with active UC. As the aforementioned genes regulate the transcription of inflammatory genes, we expected a similar disruption in the expression of the inflammatory genes. Indeed, an inflammatory gene expression analysis revealed that *IκB*, *IL10*, *NFκB1*, *NFκB2*, *IL6* and *TNFα* were significantly upregulated in patients with UC (*p* < 0.05, *p* < 0.0001, *p* < 0.05, *p* < 0.0001, *p* < 0.0001 and *p* < 0.01, respectively) compared to healthy controls (Figure 1A). These findings strengthen our hypothesis that an interaction exists between the circadian clock and the inflammatory response in patients with active UC. Analysis of the expression of genes encoding regulators of both the circadian clock and inflammation revealed that *RORγ*, *PGC1α*, *PPARα* and *PPARγ* were significantly downregulated in patients with active UC (*p* < 0.0001, *p* < 0.0001, *p* < 0.01 and *p* < 0.0001, respectively), compared to healthy controls (Figure 1A). In contrast, the expression of *RORα* was upregulated in patients with UC (*p* < 0.0001) and *REV-ERBα*, *AMPK* and *SIRT1* expression did not differ between patients and controls (*p* > 0.05) (Figure 1A). To evaluate the robustness of these results, we repeated the same analysis on a second transcription dataset, the GSE-117993 study. The analysis showed similar results for the expression levels of *BMAL1*, *PER2*, *CRY2*, *RORγ*, *PGC1α*, *PPARγ*, *IκBα*, *IL10*, *NFκB2*, *IL6* and *TNFα* in patients with active UC compared to healthy controls (Appendix A). The similar results obtained in two independent datasets suggest that our analysis uncovered a characteristic disruption in the molecular circadian clock in patients with active UC. 

### 2.2. Expression Pattern of Genes Encoding Clock, Inflammation and Their Mutual Regulators

We next analyzed the gene expression pattern among individual UC patients and healthy controls. We found a large variability in both studies (Figure 1B, Appendix A). In the GSE-109142 study, healthy controls showed a more uniform pattern of gene expression, where pro- and anti-inflammatory genes (*TNFα*, *IL10*, *IL6*, *NFκB*) showed reduced expression, and clock genes (*CLOCK*, *CRY2*, *SIRT1*, *PPARα*) showed increased expression (Figure 1B). Thus, when focusing on individual participants and comparing their gene expression patterns to one another, inflammatory genes tended to be relatively downregulated in healthy controls, while clock genes tended to be relatively upregulated. While among healthy controls there was a common expression pattern of clock, inflammatory and regulatory genes, among the UC patients a variety of expression patterns was observed (Figure 1B). Some patients with UC showed reduced expression of both core clock and inflammatory genes, some showed increased expression of these genes and some showed relatively low expression of *PER2*, *RORγ*, *PGC1α*, *PPARγ* and *AMPK*. These findings further support the notion that patients with active UC present a disruption in their molecular circadian rhythm. The extent of disruption exposes inter-individual differences among patients with UC. 

### 2.3. Correlation between Genes Encoding Clock, Inflammation and Their Mutual Regulators

We next calculated the correlation between the expression of each pair of the 20 studied genes (Figure 1C). In the GSE-109142 study, the healthy control correlation map was different from that of the UC patients. In healthy controls, the inflammatory genes *TNFα* and *NFκB2* showed a negative correlation with most clock and regulatory genes (Figure 1C). In contrast, in patients with UC, *TNFα* and *NFκB2* showed a positive correlation with most clock and regulatory genes (Figure 1C). In addition, in patients with UC, the inflammatory gene *IL6* negatively correlated with the regulatory genes *RORγ*, *PGC1α* and *PPARγ* (r = −0.36, r = −0.49, and r = −0.35, respectively). To emphasize which pairs of genes show the most prominent difference in their correlation among the control and the UC groups, we calculated the absolute difference in gene correlations between the two groups (Figure 1C). *NFκB2* paired with almost all the genes analyzed, and showed the largest absolute difference between healthy controls and UC patients (>1.00 absolute difference in correlation). More specifically, the largest differences were in the correlations between *NFκB2* and *CLOCK* (control: −0.61; UC: 0.5), *NFκB2* and *CRY2* (control: −0.82; UC: 0.56), *NFκB2* and *SIRT1* (control: −0.64; UC: 0.59), *NFκB2* and *NFκB1* (control: −0.73; UC: 0.78), *NFκB2* and *IκBα* (control: −0.42; UC: 0.72), *NFκB1* and *TNFα* (control: −0.73; UC: 0.34) and *TNFα* and *CRY2* (control: −0.82; UC: 0.21) (Figure 1C). While these pairs showed opposite expression trends in healthy controls, they had similar expression trends in patients with UC. In the GSE-117993 study, among healthy controls, the inflammatory genes *TNFα* and *IL6* showed a positive correlation (r = 0.48), but no or negative correlation with most clock and regulatory genes, i.e., a similar behavior to the one evident in the GSE-109142 dataset analysis (Appendix A). In the GSE-117993 study, among patients with UC, *PGC1α* showed a positive correlation with *PPARα* and *PPARγ* (r = 0.45 and r = 0.62, respectively), but no correlation with the other genes (Appendix A). These results emphasize pairs of genes that negatively or positively correlate in their expression among healthy controls and patients with UC.

### 2.4. Comparison between Gene Expression and Clinical, Endoscopic and Histological Scores

We next determined whether gene expression levels differed according to disease severity using clinical (PUCAI, the pediatric ulcerative colitis activity index), endoscopic (MES, the Mayo endoscopic score) and histological (the NANCY score) [31] scores in the GSE-109142 study (Figure 2). Dividing gene expression levels according to clinical disease activity revealed increased expression of *BMAL1*, *NFκB2* and *IL6*, but decreased expression of *PPARγ* and *PGC1α* in patients with active UC compared to healthy controls. The difference in gene expression levels was associated with disease severity (PUCAI: mild disease (10–35), moderate disease (36–65) and severe disease (66–85)), i.e., patients with PUCAI > 65 demonstrated higher gene expression levels of *BMAL1*, *NFκB2* and *IL6*, and lower gene expression levels of *PPARγ* and *PGC1α*, compared to patients with moderate and mild disease (Figure 2A). Similarly, dividing gene expression levels according to endoscopic severity (MES 1, 2 and 3, accordingly) revealed that the difference in gene expression levels was associated with the endoscopic severity of the disease. For example, the expression of *BMAL1*, *NFκB2* and *IL6* in patients with MES 3 was higher compared to patients with moderate or mild disease, and the expression levels of *PPARγ* and *PGC1α* were lower in patients with MES 3 compared to patients with moderate or mild disease (Figure 2B). When dividing gene expression levels according to histological severity [NANCY 1 (remission to mild disease), NANCY 2 (mild to moderate disease) and NANCY 3 + 4 (moderate to severe disease), the expression of *BMAL1*, *NFκB2* and *IL6* was significantly increased, whereas the expression of *PPARγ* and *PGC1α* was significantly decreased in patients with UC compared with healthy controls. However, the change in gene expression levels was not associated with histological severity as defined by the different NANCY score categories (Figure 2C). The association between disease severity and the aforementioned gene expression levels further reinforces the association between clock and inflammation. 

## 3. Discussion

In this study, we show for the first time simultaneous expression of clock, inflammation and their mutual regulatory genes in rectal biopsies of treatment-naïve pediatric patients with UC compared to healthy controls. More specifically, the data retrieved from patients with active UC [32] revealed high expression levels of the clock genes *BMAL1*, *CLOCK*, *PER1*, *CRY1* and *RORα* and of the pro- and anti-inflammatory genes *IκB*, *IL10*, *NFκB1*, *NFκB2*, *IL6* and *TNFα*. In contrast, patients with active UC had low expression of the genes encoding mutual regulators of both the circadian clock and inflammation, i.e., *RORγ*, *PGC1α*, *PPARα* and *PPARγ*. Similar results were found in a second dataset (GSE-117993), but these results were less pronounced, possibly since fewer UC patients were included. 

The Nuclear Factor κB (NF-κβ) signaling cascade has been shown to be a central mediator of inflammation in IBD. It activates several processes and leads to increased transcription of pro-inflammatory mediators, such as IL-6, tumor necrosis factor-α (TNF-α) and interferon (IFN)-γ [33,34]. The expression of *BMAL1* and *RORα*, a core clock gene and its positive regulator [12], was correlated in our study, and showed upregulation in patients with active UC compared to healthy controls. RORα has been shown not only to control core clock gene expression but also to control the expression of pro-inflammatory cytokines, including IL6 and IL8, through the NF-κB signaling pathway [14,15]. At the molecular level, RORα negatively interferes with NF-κB signaling by inducing the expression of its suppressor IκBα [15]. Thus, the high levels of *RORα* we found can explain the increased expression of *BMAL1* and *IκBα* in patients with UC. However, the high levels of *RORα* were not sufficient to completely interfere with NF-κB signaling [15], as demonstrated by the high levels of *NFκB1* and *NFκB2* and their downstream effectors *IL6* and *TNFα* in patients with UC. This may be due to the decreased expression of the other negative regulators of the NF-κB signaling pathway: *PPARα*, *PPARγ*, *RORγ* and *PGC1α*. 

*PPARα*, *PPARγ*, *RORγ* and *PGC1α* were correlated and showed reduced expression in patients with UC. The circadian clock controls the expression of these regulators, and, in turn, they positively regulate *BMAL1* expression [17,18,35,36] (Figure 3). In light of their low expression levels, we expected *BMAL1* to show reduced expression as well. However, the expression level of *BMAL1* was high and correlated with its classical direct positive regulator, RORα. It has been shown that the activation of PPARs, RORγ and PGC1α exerts anti-inflammatory properties, mainly by suppressing the expression of pro-inflammatory genes, such as IL-17, IL-1β and TNFα [19,20,37]. More specifically, PGC1α was found to positively regulate the expression of the anti-inflammatory cytokine IL-10 [38]. In a rat model of induced colitis, PGC1α was found to be low [39], similarly to our results. In addition, PPARγ, a negative regulator of NF-κB-dependent inflammation, was found to be reduced in colonocytes of patients with UC [40,41]. Furthermore, the activation of the AMPK-SIRT1-PGC1α pathway has also been shown to downregulate the expression of pro-inflammatory cytokines [29,39]. We observed reduced expression of *PGC1α*, but not of *AMPK* or *SIRT1*. Thus, in patients with active UC, the downregulated expression of the regulators PPARs, RORγ and PGC1α may contribute to an upregulated inflammatory response (Figure 3).

The contradictory expression patterns of the different regulators—the reduced expression of some, i.e., *RORγ*, *PGC1α*, *PPARα* and *PPARγ*, alongside the increased expression of others, i.e., *RORα*—leads to a disruptive inflammatory response in which both pro-inflammatory (*NFκB1*, *NFκB2*, *IL6* and *TNFα*) and anti-inflammatory (*IκB*, *IL10*) mediators are expressed in patients with UC. These regulators regulate clock gene expression and their expression is regulated by the circadian clock mechanism [42], which leads to a vicious cycle of complete circadian misalignment (Figure 3).

Dividing gene expression levels according to clinical, endoscopic and histological disease activity revealed that the differences in gene expression between UC and healthy controls are associated with clinical and endoscopic disease severity but not with histological severity. Mucosal healing (MH) is a therapeutic goal in UC [31]. In the same manner that fecal calprotectin serves as a surrogate marker [43] of MH and as such decreases the number of invasive endoscopies performed to monitor disease activity, clock gene expression levels may serve in the future as a non-invasive surveillance tool for monitoring disease activity in patients with IBD. Quantification of gene expression could be monitored in white blood cells and standardized according to the levels in healthy people. In addition, treatment of the circadian clock may potentially attenuate disease activity. Further studies and larger cohorts are needed to clarify the association between clock gene expression patterns and disease severity.

As mentioned above, patients with active UC presented high expression levels of core clock genes. However, this expression pattern of clock genes is inconsistent with previous findings in which lower levels of clock gene expression were found [2,3,5,44]. Patients with active inflammatory bowel disease (IBD) have been shown to have great variability in gene expression [2,3,5,44]. Indeed, when we plotted all patients with active UC and healthy controls for their individual gene expression levels, a large variability was exposed in both datasets amongst the UC patients. In contrast, the healthy controls showed quite a uniform pattern of gene expression, where the expression of both clock genes (*BMAL1*, *CLOCK*, *PER1*, *PER2*, *CRY1*) and inflammatory genes (*TNFα*, *IL10*, *IL6*, *NFκB*) was correlated and had a decreased expression. This variable expression amongst UC patients may explain the previously mentioned inconsistency. In addition, different exposures to light and sampling times within the cohorts could lead to different gene expression patterns as well.

The limitations in our study include the small number of healthy participants in the first cohort (GSE-109142) and low number of UC patients in the second cohort (GSE-117993). In addition, the database used did not specify the time of biopsy sampling, making it hard for us to relate to changes in gene expression throughout the day. 

## 4. Materials and Methods

### 4.1. Data Acquisition

Normalized and batch-corrected expression data were retrieved from the IBD Transcriptome and Metatranscriptome Meta-Analysis (TaMMA IBD) platform (version 1.0) (https://osf.io/yrxa7/) (accessed on 16 July 2016). We analyzed the transcriptomic data originally sourced from the GSE-109142 and GSE-117993 [32,45] datasets of the Gene Expression Omnibus (GEO) [46]. Normalization and batch correction were crucial for comparing the results of the two separate and independent experiments. Metadata and information on experimental conditions were retrieved from the GEO archive. Out of the entire transcriptome, which included ~6 × 10^4^ genes, we focused on circadian clock, inflammation and regulatory genes (a total of 20 genes). The genes we analyzed included clock genes (*BMAL1*, *CLOCK*, *PER1*, *PER2*, *CRY1*, *CRY2*), inflammatory genes (*IκB*, interleukin 10 (IL10), *NFκB1*, *NFκB2*, *IL6*, *TNFα*) and genes encoding regulators of both the circadian clock and inflammation (*RORα*, *RORγ*, *REV-ERBα*, *PGC1α*, *PPARα*, *PPARγ*, *AMPK*, *SIRT1*). All transcriptomes analyzed originated from rectal biopsies, obtained during diagnostic colonoscopy performed in pediatric patients. Study number GSE-109142 included 206 treatment-naïve pediatric patients with UC (age 12.9 ± 3.2; 54% male) and 20 healthy controls (age 13.9 ± 3.3; 45% male). Study number GSE-117993 included 43 treatment-naïve pediatric patients with UC (age 12.55 ± 3.49; 59.5% male) and 55 healthy controls (age 11.72 ± 3.34; 61.8% male).

### 4.2. Plotting and Data Analysis

Prior to plotting, we log2-transformed the retrieved batch-corrected and normalized counts of the 20 selected genes. Log2 transformation is essential for stabilizing variance in data, normalizing the data by reducing the influence of extremely high values, linearizing fold changes and bringing the data closer to a normal distribution to increase the statistical validity of analyses. For clustering and plotting heatmaps, we scaled and centered the transformed counts, and used complex heatmap R package [47]. We clustered the data using hierarchical clustering and Euclidean distance for identification of co-expressed genes and differentially expressed genes, for exploring associations between gene expression patterns and sample characteristics, and for visualization of the patterns in the data. We used the ggplot2 package [48] to generate jitter plots. We calculated *p*-values using the paired means difference Wilcoxon signed-rank test via ggpubr package (ggpubr: ‘ggplot2’ Based Publication Ready Plots. R package version 0.4.0, https://CRAN.R-project.org/package=ggpubr) (accessed on 10 February 2023), and corrected for multiple tests using Benjamini–Hochberg correction. The Wilcoxon test was chosen as it does not assume that the data follow a normal distribution and it is used for the analysis of small sample sizes. For plotting the Spearman’s correlation between the expression of pairs of genes among samples in each study group (e.g., UC patients and control), we used the corrplot package (R package ‘corrplot’: Visualization of a Correlation Matrix. R package version 0.92, https://github.com/taiyun/corrplot) (accessed on 31 August 2022). Spearman’s correlation was chosen as it is a non-parametric measure that does not assume a linear relationship and does not assume normality of the data, making it suitable for capturing non-linear associations. Positive correlation was determined when r > 0.3, negative correlation when r < −0.3 and no correlation when r was between −0.3 and 0. To plot the difference in these correlations between the two study groups, we subtracted the Spearman correlations of the UC group from the Spearman correlation of the controls, and plotted the absolute values of this subtraction using the complexheatmap package.

## 5. Conclusions

Herein, we found that clock genes and inflammatory genes were significantly upregulated in patients with active UC. However, the genes encoding the mutual regulators of the circadian clock and inflammation were significantly downregulated in patients with UC. The difference in gene expression levels was associated with clinical severity and endoscopic disease severity but not with histological severity. In patients with active UC, the disruption of clock genes is associated with abnormal mucosal immune response. A disrupted expression of genes encoding clock, inflammation and their mutual regulators together may play a role in active UC. Thus, the emerging data support the notion that clock disruption alters the expression of both its own regulators and those that regulate the inflammatory response. In turn, disruption in the expression of these regulators activates the inflammatory response, altogether leading to a disruptive phenotype of active inflammation in patients with UC. The inter-relations between the circadian clock and inflammation at the molecular level merit further study. 

## Figures and Tables

**Figure 1 ijms-25-01488-f001:**
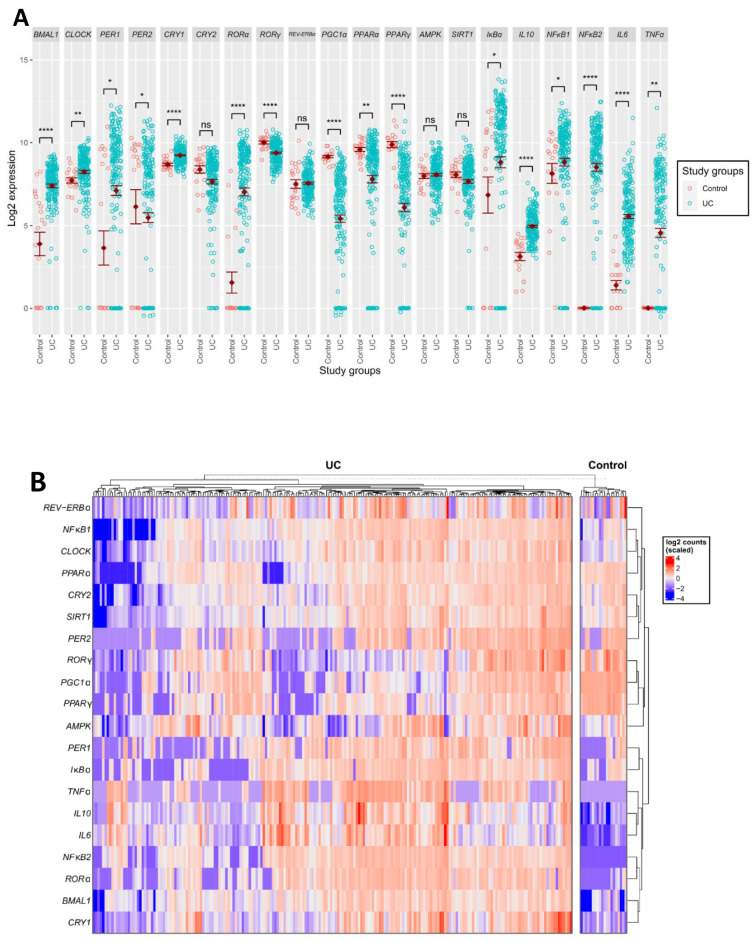
Expression and correlation of genes encoding clock, inflammation and their mutual regulators in rectal biopsies of active patients with UC: (**A**) Expression level of genes encoding clock, inflammation and their mutual regulators. Red symbols—control group. Green symbols—UC patients. ns—non-significant. (**B**) Heatmap of gene expression of healthy control vs. active patients with UC. (**C**) Correlation of gene expression between pairs of genes of healthy controls and UC patients. The far-right panel is the absolute difference between the gene pair correlation of controls and UC patients. Data are means ± SE. Asterisks denote significant differences: * is 0.05 ≥ *p* ≥ 0.01; ** is 0.01 ≥ *p* ≥ 0.001; **** *p* ≤ 0.0001.

**Figure 2 ijms-25-01488-f002:**
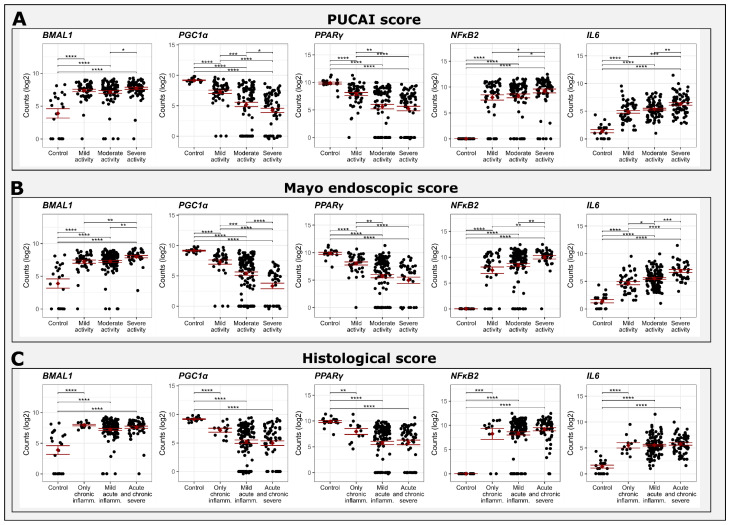
Trends of clock (*BMAL1*), inflammation (*NFkB2*, *IL6*) and mutual regulatory (*PCG1α, PPARγ*) gene expression according to clinical score (**A**), endoscopic score (**B**) and histological severity (**C**) in patients with UC. Data are means ± SE. Asterisks denote significant differences: * is 0.05 ≥ *p* ≥ 0.01, ** is 0.01 ≥ *p* ≥ 0.001, *** is 0.001 ≥ *p* ≥ 0.0001, **** *p* ≤ 0.0001.

**Figure 3 ijms-25-01488-f003:**
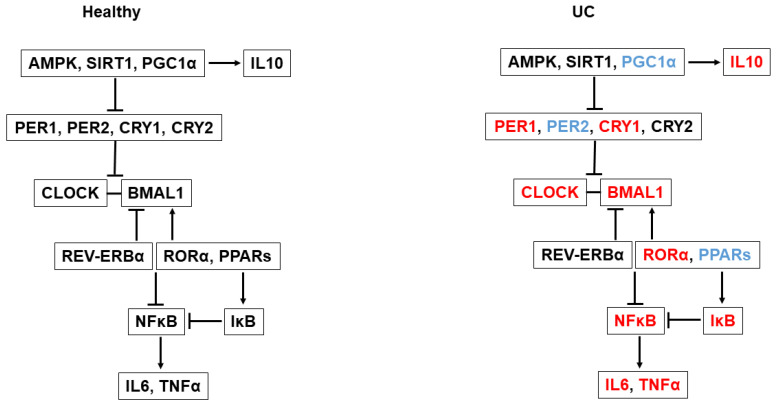
A model summarizing the relationship between the clock mechanism and inflammation in healthy subjects and UC patients. Red—increased expression; blue—reduced expression; black—unchanged expression in patients with UC compared to healthy controls.

## Data Availability

The original contributions presented in the study are included in the article’s Appendix A. Further inquiries can be directed to the corresponding author.

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
