# Peer review of "Mucosal Genes Encoding Clock, Inflammation and Their Mutual Regulators Are Disrupted in Pediatric Patients with Active Ulcerative Colitis"

_ijms, 2024, doi:10.3390/ijms25031488_

Round 1

Reviewer 1 Report

Comments and Suggestions for Authors

In this study, Labes et al. investigated the circadian clock and inflammatory gene expression as well as the expression of genes encoding their mutual regulators in rectal biopsies of pediatric patients with active ulcerative colitis, upon diagnosis, prior to medical treatment. For this purpose, the Authors applied the Inflammatory Bowel Disease Transcriptome and Meta-transcriptome Meta-Analysis (IBD TaMMA) platform and R algorithm. 

A few comments arose during the revision that should be addressed:

  1. Authors should provide a more detailed Introduction and Methods section.
  2. Authors describe the statistical tests used to analyze the data, but they do not provide a detailed explanation of why these tests were chosen.
  3. The experimental data are discussed and presented in a very poor way and should be improved.
  4. Authors should incorporate pertinent references at the end of sentences on lines 38-41.
  5. The captions for the figures lack sufficient detail. 
  6. Figure 1 consists of a collection of sub-figures. Please insert the appropriate letters for each subfigure. 
  7. Enhance panel 1c resolution.
  8. The conclusions are overly succinct.
  9. The manuscript's font is irregular along the text.
  10. As highlighted by the Authorsthe study's limitations include the small number of healthy participants in the first cohort and the low number of ulcerative colitis patients in the second cohort. In addition, the database used does not specify the time of biopsy sampling, making it hard to relate to changes in gene expression throughout the day.

Author Response

In this study, Labes et al. investigated the circadian clock and inflammatory gene expression as well as the expression of genes encoding their mutual regulators in rectal biopsies of pediatric patients with active ulcerative colitis, upon diagnosis, prior to medical treatment. For this purpose, the Authors applied the Inflammatory Bowel Disease Transcriptome and Meta-transcriptome Meta-Analysis (IBD TaMMA) platform and R algorithm. 

A few comments arose during the revision that should be addressed:

  1. Authors should provide a more detailed Introduction and Methods section.

We thank the reviewer for this comment. We have provided more details in the Introduction and Methods sections.

  1. Authors describe the statistical tests used to analyze the data, but they do not provide a detailed explanation of why these tests were chosen.

We thank the reviewer for this comment. We have provided a detailed explanation why the statistical tests were chosen in the Methods section.

  1. The experimental data are discussed and presented in a very poor way and should be improved.

We thank the reviewer for this comment. The comparisons and correlations we performed generated a large amount of data. These data are available in the figures and supplementary figures. We could not describe in full all comparisons and correlations in the Results section. We summarized the overall trends and highlighted the major differences. As all the data exists in the figures and supplementary data, readers can easily find any comparison and correlation that may be of interest to them.

  1. Authors should incorporate pertinent references at the end of sentences on lines 38-41.

We thank the reviewer for this comment. We have added a reference at the end of sentences on lines 38-41.

  1. The captions for the figures lack sufficient detail. 

We thank the reviewer for this comment. We have elaborated the captions for the figures.

  1. Figure 1 consists of a collection of sub-figures. Please insert the appropriate letters for each subfigure. 

We thank the reviewer for this comment. We have inserted the appropriate letters for each panel in Figure 1.

  1. Enhance panel 1c resolution.

We thank the reviewer for this comment. We have enhanced the resolution of panel 1c.

  1. The conclusions are overly succinct.

We thank the reviewer for this comment. We have extended the conclusions section.

  1. The manuscript's font is irregular along the text.

We thank the reviewer for this comment. The font has been corrected throughout the manuscript.

  1. As highlighted by the Authors, the study's limitations include the small number of healthy participants in the first cohort and the low number of ulcerative colitis patients in the second cohort. In addition, the database used does not specify the time of biopsy sampling, making it hard to relate to changes in gene expression throughout the day.

These limitations are mentioned at the end of the manuscript.

Reviewer 2 Report

Comments and Suggestions for Authors

In this manuscript the authors describe interesting results demonstrating a different expression pattern of genes encoding clock, inflammation and their mutual regulators,  in pediatric patients with Ulcerative Colitis in comparison with control subjects. Also, changes in gene expression were associated with clinical and endoscopic score.

The manuscript is well written and the description and presentation of data are clear and attractive, thus it could arouse the readers’ interest.

In my opinion the manuscript is suitable for publication after minor revision.

 1. In figure 1 the letters a, b and c are missing.

 2. The authors hypothesize that clock “gene expression levels” may serve as a surveillance tool for monitoring disease activity in patients with IBD. The authors should clarify how they think the quantification of gene expression could be standardized and how reference values could be established.

3. Could the analysis of some mentioned proteins in blood be hypothesized to validate their diagnostic or prognostic value? In this case it may be easier to establish reference values.

Author Response

In this manuscript the authors describe interesting results demonstrating a different expression pattern of genes encoding clock, inflammation and their mutual regulators, in pediatric patients with Ulcerative Colitis in comparison with control subjects. Also, changes in gene expression were associated with clinical and endoscopic score.

The manuscript is well written and the description and presentation of data are clear and attractive, thus it could arouse the readers’ interest.

In my opinion the manuscript is suitable for publication after minor revision.

  1. In figure 1 the letters a, b and c are missing.

We thank the reviewer for this comment. We have added the letters in Fig. 1.

  1. The authors hypothesize that clock “gene expression levels” may serve as a surveillance tool for monitoring disease activity in patients with IBD. The authors should clarify how they think the quantification of gene expression could be standardized and how reference values could be established.

We thank the reviewer for this comment. We have added a possible way to quantify gene expression and how reference values could be established in the Discussion section..

  1. Could the analysis of some mentioned proteins in blood be hypothesized to validate their diagnostic or prognostic value? In this case it may be easier to establish reference values.

The genes analysed in this study all encode intracellular rather than secreted proteins. Therefore, they cannot be measured in blood. We can measure their gene expression in white blood cells. We have mentioned this point in the Discussion section.

Round 2

Reviewer 1 Report

Comments and Suggestions for Authors

The revised manuscript is now suitable for publication.